# The Molecular Epidemiology of Pneumococcal Strains Isolated from the Nasopharynx of Preschool Children 3 Years after the Introduction of the PCV Vaccination Program in Poland

**DOI:** 10.3390/ijms24097883

**Published:** 2023-04-26

**Authors:** Karolina Kielbik, Ewelina Grywalska, Andrzej Glowniak, Grażyna Mielnik-Niedzielska, Izabela Korona-Glowniak

**Affiliations:** 1Department of Pharmaceutical Microbiology, Faculty of Pharmacy, Medical University of Lublin, 20-093 Lublin, Poland; 2Department of Clinical Immunology, Faculty of Medicine, Medical University of Lublin, 20-093 Lublin, Poland; ewelina.grywalska@umlub.pl; 3Department of Cardiology, Medical University of Lublin, 20-093 Lublin, Poland; andrzej.glowniak@umlub.pl; 4Department of Pediatric Otolaryngology, Phoniatrics and Audiology, Medical University of Lublin, 20-093 Lublin, Poland; grazyna.mielnik-niedzielska@umlub.pl

**Keywords:** *Streptococcus pneumoniae*, resistance genes, MLST, mobile genetic elements, pili islets

## Abstract

The genetic mechanisms of resistance, clonal composition, and the occurrence of pili were analyzed in 39 pneumococcal strains isolated from healthy children in the southeastern region of Poland. Strains with resistance to combinations of erythromycin, clindamycin, and tetracycline were found in clonal groups (CGs) related to Tennessee 23F-4 and Taiwan 19F-14 clones. Capsular switching possibly occurred in the Spain 9V-3 clone and its variants to serotypes 35B and 6A, as well as DLVs of Tennessee 23F-4 to serotype 23A. The double-locus variants of Colombia 23F-26 presented serotype 23B. The major transposons carrying the erythromycin and tetracycline resistance genes were Tn*6002* (66.6%), followed by Tn*916* (22.2%) and Tn*2009* (11.1%). The macrolide efflux genetic assembly (MEGA) element was found in 41.7% of all erythromycin-resistant isolates. The majority of the isolates carrying the PI-1 gene belonged to the CGs related to the Spain 9V-3 clone expressing serotypes 35B and 6A, and the presence of both PI-1 and PI-2 was identified in CG4 consisting of the isolates related to the Taiwan 19F-14 clone expressing serotypes 19F and 19A. Importantly, in the nearest future, the piliated strains of serogroups 23B, 23A, and 35B may be of concern, being a possible origin of the emerging clones of piliated non-vaccine pneumococcal serotypes in Poland. This study reveals that nasopharyngeal carriage in children is an important reservoir for the selection and spreading of new drug-resistant pneumococcal clones in the community after the elimination of vaccine serotypes.

## 1. Introduction

*Streptococcus pneumoniae* is one of the major bacterial pathogens colonizing the nasopharynx of young children, mainly asymptomatically. Colonized children have been found to be the main reservoir of pneumococci, playing a key role in the dissemination and selection of multidrug-resistant strains [1]. Moreover, pneumococci are essential etiological agents of pneumonia, bacteremia, meningitis, and acute otitis media in both children and adults [2]. The treatment of pneumococcal infections has become challenging due to the spreading of penicillin-resistant strains and the rapid development of resistance to other antibiotics, including macrolides.

*S. pneumoniae* is armed with several virulence factors, including capsular polysaccharide, the main virulence factor, and many proteins involved in the development of pneumococcal disease, namely pneumolysin (Ply), the major autolysin (LytA), two neuraminidases (NanA and NanB), choline-binding protein A (CbpA), hyaluronate lyase (Hyl), and the pili, i.e., multimetric filaments [3,4]. Pneumococcal pathogenesis is based on these proteins and enzymes, which have distinct roles in different host niches and stages of infection [3,4]. Pneumococci may possess two different pilus islets (PIs) that encode the structural genes necessary for the biosynthesis of two antigenically different types of pili, PI-1 and PI-2. These surface-exposed hair-like structures favor nasopharyngeal colonization and invasion [5].

Poland is one of the last countries in Europe that introduced mandatory anti-pneumococcal vaccinations at no charge. In 2017, the 10-valent pneumococcal conjugate vaccine (PCV-10) was added to the childhood vaccination schedule for all children born that year without follow-up. In the preceding 10-year period, PCVs were recommended to all children for a fee and free of charge to high-risk groups. The 13-valent pneumococcal conjugate vaccine (PCV-13) was then available for personal requirements and was often used [6].

In our previous studies carried out in the prevaccination period in Poland in 2002–2003, the rate of *S. pneumoniae* carriage ranged from 33% to 44% in healthy preschool children [7,8] and in the 2011–2012 period, in children with recurrent upper respiratory tract infections, the observed rate of carriage ranged from 64.1% to 70.2% [9,10]. In a recent study completed in 2020 (three years after the introduction of mandatory PCV), the colonization of *S. pneumoniae* in healthy children aged 1–6 years was halved to 23.3%. However, due to the replacement with non-vaccine serotypes, there was no difference in carriage prevalence in vaccinated and unvaccinated children [11]. The introduction of PCVs is expected to cause serotype replacement and alters the genetic composition of pneumococci in response to vaccine-induced selective pressure [12,13]. Considering the high recombination rates in pneumococci [14], the determination of their genetic diversity and serotype expression, especially antibiotic-resistant strains, can be useful to monitor the evolution of this bacterium under antibiotic and vaccine pressure. The distribution of pneumococcal serotypes and genotypes varies among countries [15].

Multilocus sequence typing (MLST) is considered the standard method for the investigation of genetic diversity among isolates and the epidemiological surveillance of *S. pneumoniae* from different continents [16].

Multidrug resistance in *S. pneumoniae* strains, mainly resistance to tetracycline encoded by *tet*(M), macrolides encoded by *erm*(B) and/or *mef*(A/E), and chloramphenicol encoded by *cat*, is generally associated with possessing mobile genetic elements, including Tn*916* and Tn*5252* families, as well as their unique ability to transformation and recombination resulting from genetic plasticity [17]. Pneumococcal resistance to erythromycin and tetracycline is related to the insertion of the resistance gene *erm*(B) or *mef*(A/E) (or both) into the transposons of the Tn*916* family forming Tn*6002*, Tn*1545* (also carrying the kanamycin resistance gene *aph3*-*III*), and Tn*3872* (carrying *erm*(B) within Tn*917* and transposase genes *tnpA* and *tnpR*) [17]. Moreover, two composite elements of the Tn*916* family, containing *tet*(M) plus MEGA (macrolide efflux genetic assembly) carrying the *mef*(E) gene (Tn*2009*) and *tet*(M), *erm*(B), and MEGA (Tn*2010*) have been described [18,19].

The aim of this study was to characterize the *S. pneumoniae* strains detected in preschool children 3 years after the introduction of the vaccination program in Poland. Serotypes; antimicrobial susceptibility patterns; genotypic characteristics, including the transposons with resistance genes; and pilus prevalence were analyzed, followed by the clonality analysis of the isolates performed using the MLST method. Moreover, we investigated the prevalence of pilus in pneumococcal isolates and their association with serotypes and antibiotic resistance.

## 2. Results

The 39 pneumococcal strains isolated from 176 healthy children aged 1–6 years (prevalence—22.2%) were characterized for genotypic properties. They were identified with the PCR detection of *lytA* and *ply* genes, and all of them were identified as capsular with the detection of the *cpsA* gene. However, only 28.2% of the isolates (n = 11) amplified the targets for PI-1 and 28.2% (n = 11) for PI-2. Six strains expressed both pilus islets (15.4%). Serotype determination and antibiotic resistance patterns of these strains were presented previously [20]. They were from 10 different serotypes. Notably, 17 out of the 39 pneumococcal isolates (43.6%) were susceptible to all the tested antimicrobial agents. They belonged to serotype 23A (five isolates), serotype 23B (four isolates), 35F/47F (three isolates), 10A (two isolates), and 6A, 15B, and 35B (one isolate per each serotype).

### 2.1. Resistance Genes and Presumptive Transposons

Of the 39 isolates, 8 (20.5%) were not susceptible to penicillin (Figure 1). Resistance to erythromycin, clindamycin, tetracycline, and cotrimoxazole was 30.7% (12 isolates), 15.4% (6 isolates), 23.1% (9 isolates), and 25.6% (10 isolates), respectively. Stratifying the tested population by vaccination status, 7 out of 28 (25%) isolates were collected from vaccinated children, and 5 of the 11 (45%) isolates collected from unvaccinated children were resistant to erythromycin (*p* = 0.26). There were no significant differences in resistance to penicillin and co-trimoxazole between these groups (21% vs. 18%, *p* = 1.0; and 25% vs. 27%, *p* = 1.0, respectively). Interestingly, resistance to tetracycline was significantly higher among the isolates obtained from unvaccinated children than the isolates from the vaccinated group (55% vs. 11%, RR 5.1, 95%CI 1.5–16.9, *p* = 0.0078).

All tetracycline-resistant isolates were *tet*(M)-gene-positive. The analyses of erythromycin-resistant strains revealed that 50% of the isolates were resistant to macrolide lincosamide–streptogramin B (MLS_B_) with a constitutive phenotype. These isolates harbored the *erm*(B) gene only. The macrolide (M) resistance phenotype was observed in 50% of the isolates and harbored the *mef*(*E*) gene (Table 1). None of the isolates possessed both macrolide resistance genes.

The most frequent transposon carrying the erythromycin and tetracycline resistance genes was Tn*6002* (66.6%), followed by Tn*916* (22.2%) and Tn*2009* (11.1%). The MEGA element was found in 41.7% of all erythromycin-resistant isolates (Table 1). Stratifying our population by vaccination status, in 50% of the erythromycin-resistant isolates collected from unvaccinated children, 66.7% and 33.3% had *erm*(B) and *mef*(E) genes, respectively. Amongst the erythromycin-resistant isolates collected from vaccinated children, the percentage was the opposite (33.3% and 66.7% isolates with *erm*(B) and *mef*(E), respectively). The same prevalence was observed with the mobile elements containing these genes, namely Tn*916* family transposons with *erm*(B) and *tet*(M) genes and the MEGA element not integrated into transposon from the Tn*916* group. Out of the eight pneumococcal strains resistant to erythromycin and/or tetracycline isolated from unvaccinated children, 75% transposons were detected from the Tn*916* family (Tn*916*—25% and Tn*6002*—50%), and 25% of the strains possessed the MEGA element. In the resistant strains isolated from vaccinated children, 50% possessed transposon from the Tn*916* family (Tn*6002* and Tn*2009*), and 50% possessed the MEGA element in the pneumococcal chromosome backbone (Table 1).

### 2.2. Multilocus Sequence Typing (MLST) and PMEN Clones

The 39 nasopharyngeal isolates were assigned to 28 STs, 2 of which were not previously reported in the MLST database (ST17901 and ST17907, Table 1). MLST revealed six clonal groups (CGs). Six STs consisted of more than one isolate, and twenty-two STs were identified with one single isolate. ST1349 (n  =  6) and ST15380 (n  =  3) were most frequently detected. We observed a correlation between sequence type distribution with the serotype.

Notably, 18 strains were closely related to 7 of the 43 Pneumococcal Molecular Epidemiology Network (PMEN) clones (https://www.pneumogen.net/gps/pmen.html, accessed on 3 March 2023). A total of four isolates had identical or a single-locus variant (SLV) compared with the international clone Spain 9V-3. Fourteen strains were double-locus variants (DLVs) of six other PMEN clones (Tennessee23F-4, Taiwan 19F-14, Colombia 23F-26, Netherlands 8-33, Poland 6B-20, and Greece 21-30).

Presumably, capsular switching cases occurred. The Spain 9V-3 ST156 clone with resistance pattern P was similar to the isolates with serotype 35B ST156 and resistance pattern PE and variants with serotype 35B (ST44 PE) and 6A (ST162 S). Similarly, the DLVs of Tennessee 23F-4 presented serotype 23A, the DLVs of Colombia 23F-26 presented serotype 23B, and the DLVs of Poland 6B-20 presented serotype 6A (Table 1).

Resistance to various antimicrobials was significantly associated with PMEN clones (Table 1). The isolates with resistance to erythromycin, clindamycin, and tetracycline were found in CGs related to Tennessee 23F-4 and Taiwan 19F-14 PMEN clones. In CG2 related to the Spain 9V-3 clone, MEGA elements were found. STs that mutually had at least six of the seven allelic variants were composed of clonal groups (CGs). A minimum spanning tree (MST) based on MLST was plotted using PHYLOVIZ 2.0. Figure 2 shows the relatedness of different serotypes with the identified STs.

## 3. Discussion

This study is the first report describing the genotypic analysis of both antibiotic-susceptible and -resistant pneumococcal carriage nasopharyngeal isolates from one region in Poland 3 years after the introduction of the vaccination program in Poland. Even though PCV vaccination has lowered the carriage of pneumococcal vaccine types, the carriage of non-vaccine types has increased due to serotype replacement [20]. Moreover, *S. pneumoniae* epidemiology is complex due to the diversity of the circulating genotypes often masked by varying capsular types. It is very likely that within the nasopharyngeal niche, the competition between different pneumococcal serotypes promotes bacterial colonization.

MLST data using housekeeping genes sequencing showed high diversity among the isolates tested (Simpson’s ID 0.970, 95%CI 0.941–0.999). Among the 28 identified STs, 6 clonal groups and 8 singletons were identified. According to the BURST analysis results, 28.2% of nasopharyngeal isolates were closely related to 7 of the 43 PMEN clones. Our results suggest that PMEN clones are not widely distributed within carriage isolates in the post-vaccination era in the region. In the studies carried out in this region before the introduction of vaccination, 58.4% of the isolates detected in children with recurrent respiratory tract infections were related to STs belonging to 17 out of the 43 resistant PMEN clones, i.e., Spain 9V-3, Spain 23F-1, Norway NT-42, and Poland 6B-20, as well as to 5 STs that were single-locus variants of PMEN-resistant clones (England 14–9, Spain 9V-3, Spain 23F-1, Greece 21–30, and Denmark 14–32) [21]. Among the pneumococcal strains isolated from the middle ear fluid taken from children with acute otitis media, 56% of the isolates belonged to the STs of the pandemic PMEN clones or their variants (SLVs and DLVs), including Spain 9V-3-ST156, England 14–9-ST9, Spain 23F-1-ST81, and Sweden 15A-25-ST63-SVL [22]. Most of these pandemic clones or their SLVs or DLVs were also detected in this study; however, they expressed distinct serotypes.

We found the strains grouped in CGs related to PMEN clones, which presented different serotypes (Spain 9V-3 clone to 35B serotype and Tennessee 23F-4 clone to 23A serotype). The competence of *S. pneumoniae* to switch its polysaccharide capsule causes the replacement of non-vaccine serotypes. An opportunity for the exchange of DNA between the types occurs when multiple serotypes of *S. pneumoniae* are concurrently carried in the nasopharynx. Serotype replacement facilitates the spread of drug resistance among diverse capsular types and allows the evasion of serotype-specific host immune defenses against a limited range of vaccine serotypes [23].

Pneumococcal macrolide resistance was mediated by two major mechanisms: a ribosomal methylase encoded by the *erm*(B) gene, causing target modification, and drug efflux via the membrane efflux pump encoded by the *mef*(A/E) gene. In our previous studies, macrolide-resistant pneumococci were mainly associated with the MLS_B_ phenotype, confirmed by the presence of the *erm*(B) gene, and none of the previously tested strains possessed the M phenotype and the *mef* gene as a sole determinant of erythromycin resistance [21,24]. In the study by Izdebski et al. [25], 4.7% of the clinical pneumococcal strains resistant to macrolides possessed the *mef*(E) gene and the M phenotype. In this study, surprisingly, 50% of the strains resistant to erythromycin presented the M phenotype.

Pneumococcal resistance to erythromycin is frequently associated with the insertion of the *erm*(B)/*mef* gene (as Tn*917* or MEGA) into the transposons of the Tn*916* family, including Tn*6002*, Tn*3872*, Tn*2009*, Tn*2010*, Tn*1545*, and Tn*6003* [17,18,19]. Geographical differences in the prevalence and spread of transposons carrying the *erm*(B) gene in pneumococci have often been reported. In European countries, Tn*6002* and Tn*3872* were the most frequent transposons detected in *S. pneumoniae* strains, whereas Tn*917* and Tn*2010* were most often observed in China and Japan [21,26,27,28,29]. The presence of Tn*916*, which is a prototype of the family, was observed in two tetracycline-resistant strains with serotype 10A, comprising 22.2% of resistant strains, significantly more than that observed in our other studies, with approximately 11% of prevalence [21,24].

However, past reports from Poland showed a 38.8% prevalence of Tn*916* [25]. In our study, Tn*6002* was found to be dominant in the tested population, with 42.9% of the strains resistant to macrolide and/or tetracycline, followed by the MEGA element carrier (35.7% resistant strains). One *mef* gene on the MEGA element was incorporated into Tn*916*, forming Tn*2009*. Surprisingly, Tn*3872*, Tn*2010*, and Tn*6003*/Tn*1545* were not detected in any of the tested pneumococci, which were very prevalent transposons from the Tn*916* family in our previous studies carried out in the prevaccination era [21,26]. However, the changes in the serotype distribution after the implementation of routine vaccination had an impact on transposon distribution. Connections between serotype 6B and Tn*3872* prevalence, as well as Tn*6002* and serotype 19F, were observed, while among Tn*6003*/Tn*1545* carriers, most either had serotype 23F or were non-typeable [21]. With vaccination, these serotypes were mostly eliminated in our population. The Tn*6002* detected in this study was observed in the strains related to PMEN clones Taiwan 19F-14, Poland 6B-20, and Tennessee 23F-4, albeit with replaced serotypes. This is meaningful in that, in this study, the *erm*(B) and *tet*(M) genes carried by the elements belonging to the Tn*916* family were mainly detected in the isolates collected from unvaccinated children.

The MEGA element can be integrated into at least four loci in the pneumococcal chromosome or can integrate into *orf6* of Tn*916* named Tn*2009* [28]. In the majority of the strains with the M phenotype in this study, the MEGA element was located in the chromosome backbone, and one strain carried Tn*2009*.

Pili PI-1 and PI-2 facilitate colonization and growth on mucosal surfaces and were found in the strains belonging to clonal types that are frequently found in the nasopharynx. PI-1 promotes colonization and contributes to epithelial adhesion, which is also a role of PI-2 [5]. Before routine pneumococcal vaccination, the genes of PI-1 and/or PI-2 in pneumococci isolated from children were mainly found in the isolates of vaccine serotypes belonging to multidrug-resistant international clones, including PNSP [5,30,31]. Vaccination decreased the rates of piliated pneumococci in invasive pneumococcal disease [32,33]. However, increased multidrug resistance among non-PCV-13 serotypes and in isolates with pilus islets was reported in a Japanese study [34]. Moreover, piliated non-vaccine-type strains emerged in East Jerusalem possibly due to the intrinsic advantage of PI-1 for colonization [35]. In this study, the vast majority of the isolates carrying PI-1 genes belonged to CGs related to the Spain 9V-3 clone expressing serotype 35B and 6A, and the presence of both PI-1 and PI-2 was identified in CG4, consisting of the isolates related to the Taiwan 19F-14 clone expressing serotype 19F and 19A. It can be expected that piliated pneumococci will become rare if vaccination eradicates the vaccine types in healthy children. Importantly, in the nearest future, the piliated strains of serogroups 23B, 23A, and 35B may be of concern, as they are the possible origins of the emerging clones of piliated non-vaccine pneumococcal serotypes in Poland. Due to the role of PI-1 and PI-2 in colonization promotion and epithelial adhesion contribution, they have often been found in successful pneumococcal clones, i.e., PMEN clones. Therefore, it can be expected that the prolonged colonization of non-vaccine serotypes facilitated by pili exposes the isolates to antibiotic selection [5,35].

This study has a limitation. Our sample size was not sufficiently large to have the power to exemplify the situation in the whole of Poland. Nevertheless, the strength of this study lies in performing repeated observations in the same regions with the use of a comparable methodology. A longitudinal carriage survey performed for the same population should allow for a more accurate assessment of the vaccine impact over time.

## 4. Materials and Methods

### 4.1. Bacterial Strains

A total of 39 strains from the previously described collection, isolated from the nasopharynx of 176 healthy children between 1 and 6 years old and belonging to different social groups in Lublin Voivodeship in Poland, were tested [20].

### 4.2. Detection of Resistance Genes, Transposons, and Pili

Chromosomal DNA was extracted from an overnight culture of the isolates grown on blood agar using a QIAamp DNA Mini Kit (Germantown, MD, USA), following the manufacturer’s recommendations. PCR experiments with the primer pairs and the protocols described previously [23] were used to detect the erythromycin resistance gene *mef*(E) and the tetracycline resistance gene *tet*(M). The PCR product of the *mef*(A/E) gene was digested with *Bam*HI (Fermentas, Thermo Fisher Scientific, Waltham, Massachusetts, USA) to differentiate between the *mef*(A) and *mef*(E) gene subclasses. The Tn*916* and Tn*917* transposon-related genes (*int*, *xis*, *tnpA*, and *tnpR*), the tetracycline-resistance gene *tet*(M), and the promoter of the *aph3*′-*III* gene were detected with PCR, using the primers described in previous studies [7,11,20,23,33]. The resistance gene combinations related to the different presumed transposons were Tn*6002* (*erm*(B), *tet*(M), *int*, and *xis*); Tn*2010* (*erm*(B), *tet*(M), *int*, *xis*, and *mef*(E)); Tn*3872* (*erm*(B), *tet*(M), *tnpA*, and *tnpR*); Tn*1545*; or Tn*6003* (*erm*(B), *tet*(M), *int*, *xis*, and *aph3*′-*III*). *S. pneumoniae* virulence genes (*lytA*, *ply*, and *cpsA*) were detected using published primers [21]. The primers targeting *rrgA* (3′-end) and *rrgB* (5′-end) amplified PI-1, while those targeting *pitA* (3′-end) and *pitB* (5′-end) amplified PI-2. To confirm the absence of these genes, we used the primers that targeted the flanking regions [30].

### 4.3. Multilocus Sequence Typing (MLST)

The PCR amplicons of the housekeeping genes *aroE*, *gdh*, *gki*, *recP*, *spi*, *xpt*, and *ddl* were sequenced [36], and allele sequences of each of the seven loci were identified in the database of the pneumococcal multilocus sequence typing (MLST) website (http://spneumoniae.mlst.net, accessed on 24 April 2023). A unique combination of alleles was identified for each isolate as a sequence type (ST). New allelic numbers or new ST numbers were assigned by the curator of the pneumococcal MLST website. The discriminatory power of the MLST method was calculated using Simpson’s diversity index (SID) with the online tool available at http://www.comparingpartitions.info (accessed on 12 April 2023).

Phylogenic tree visualization was performed with the use of the PHYLOViZ program to generate minimum spanning and neighbor-joining trees (https://online.phyloviz.net/, accessed on 6 March 2023).

### 4.4. Cluster Analysis and PMEN Database

BURST, an algorithm used to group MLST-type data based on a count of the number of profiles that match each other at specified numbers of loci (https://pubmlst.org/bigsdb?db=pubmlst_spneumoniae_isolates&page=plugin&name=BURST, accessed on 25 February 2023), was used to investigate the relationships between the isolates based on the stringent group definition of six out of the seven shared alleles. Single-locus variants (SLVs) and double-locus variants (DLVs) were considered sufficiently related to be members of the same clonal group of the parental ST clone (based upon related sequence types (BURST analysis)). Furthermore, STs were compared with data in the Pneumococcal Molecular Epidemiology Network (PMEN) database (https://www.pneumogen.net/gps/pmen.html, accessed on 3 March 2023).

### 4.5. Statistical Analysis

The statistical analysis was performed with GraphPad Prism 9.Ink (GraphPad Software, LCC, San Diego, CA, USA). Categorical variables were analyzed using Fisher’s exact tests. *p*-values  <  0.05 were considered significant.

## 5. Conclusions

In conclusion, this study presents a thorough molecular analysis of *S. pneumoniae* isolates circulating among preschool children in southeastern Poland. The prevalent serotypes were 23B, 23A, 6A, 10A, and 35B. A total of 28.2% of nasopharyngeal isolates were closely related to 7 out of the 43 PMEN clones. The distribution of transposons resulted in clear genetic diversity and different serotype prevalence. This is meaningful in that the elements belonging to the Tn*916* family were mainly detected in the isolates collected from unvaccinated children. Such epidemiological studies are necessary to investigate the effect of current pneumococcal conjugate vaccines (PCVs) on serotype distribution and antibiotic resistance among *S. pneumoniae* isolates, including the increase in non-vaccine serotypes, especially those resistant to antibiotics. The replacement of serotypes after PCVs has been an issue due to geographical discrepancies worldwide. Nasopharyngeal carriage in preschool children seems to be an important reservoir for the selection and dissemination of new drug-resistant pneumococcal clones in the community after the elimination of vaccine serotypes.

## Figures and Tables

**Figure 1 ijms-24-07883-f001:**
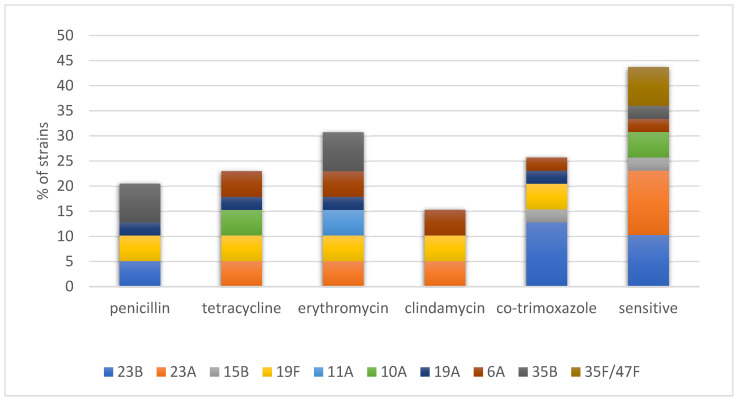
Serotype distribution between drug-resistant and sensitive nasopharyngeal *S. pneumoniae* isolates from healthy children.

**Figure 2 ijms-24-07883-f002:**
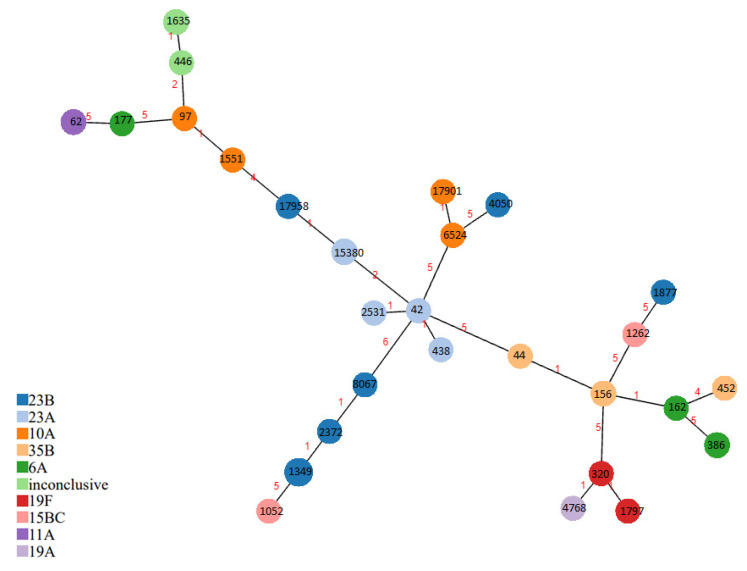
Minimum spanning tree based on allelic profiles for nasopharyngeal *S. pneumoniae* isolates from healthy children. Colors show the presence of different serotypes (inconclusive—serotype 35F/47F). Circles present the sequence types (STs) with their size corresponding to the number of isolates. Link labels are shown in absolute count of differences.

**Table 1 ijms-24-07883-t001:** Antibiotic resistance patterns, capsular types, and molecular characterization of nasopharyngeal *S. pneumoniae* isolates from healthy children.

Clonal Group	Sequence Types (n)	Associated PMEN Clone	Serotype (n)	Resistance Pattern	Resistance Genes (n)	Transposon Genes	Presumptive Transposon	Pili
1	42 * (1)		23A	ECCTe	*erm*(B), *tet*(M)	*Int*/*xis916*	Tn*6002*	
	438 (1)	Tennessee23F-4, DLV	23A	S				
	2531 (1)	Tennessee23F-4, DLV	23A	ECCTe	*erm*(*B*), *tet(M)*	*Int/xis916*, *orf20-19*(3.7 kb)	Tn*6002*	
	15380 (3)		23A (3)	S (3)				PI-1 + PI-2 (1)
	17958 (1)		23A	S				PI-2
2	44 (1)	Spain 9V-3, SVL	35B	PE	*mef*(E)	*mel*	MEGA	PI-1
	156 * (2)	Spain 9V-3	35B (2)	PE (2)	*mef*(E) (2)	*mel*	MEGA (2)	PI-1 (2)
	162 (1)	Spain 9V-3, SVL	6A	S				PI-1
3	97 (1)		10A	S				
	446 (1)		35F/47F	S				
	1551 (1)		10A	S				
	1635 (2)		35F/47F (2)	S (2)				
4	320 * (1)		19F	PECCTeSXT	*erm*(B), *tet*(M)	*int/xis916*, *orf20-19*(3.7 kb)	Tn*6002*	PI-1 + PI-2
	4768 (1)		19A	PETeSXT	*mef*(E), *tet*(M)	*int/xis916*, *orf20-29*(3.7 kb)	Tn*2009*	PI-1 + PI-2
	17907 (1)	Taiwan 19F-14, DLV	19F	PECCTeSXT	*erm*(B), *tet*(M)	*int/xis916*, *orf20-19*(3.7 kb)	Tn*6002*	PI-1 + PI-2
5	1349 (6)	Colombia 23F-26, DLV	23B (6)	SXT (4) S (2)				PI-1 (1), PI-2 (1)
	2372 * (1)		23B	P				PI-1 + PI-2
	8067 (1)		23B	PSXT				PI-2
6	6524 (1)		10A	Te	*tet*(M)	*int*/*xis916*, *orf20-19*(0.8kb)	Tn*916*	
	17901 (1)		10A	Te	*tet*(M)	*int*/*xis916*, *orf20-19*(0.8kb)	Tn*916*	
Singletons	62 (2)	Netherlands 8-33, DLV	11A (2)	E (2)	*mef*(E) (2)	*mel*	MEGA(2)	PI-2
	177 (1)		6A	SXT				PI-1 + PI-2
	386 (2)	Poland 6B-20, DLV	6A (2)	ECCTe (2)	*erm*(B), *tet*(M) (2)	*int*/*xis916*, *orf20-19*(3.7 kb)	Tn*6002* (2)	
	452 (1)		35B	S				
	1025 (1)		15B	SXT				
	1262 (1)		15B	S				PI-2
	1877 (1)	Greece 21-30, DLV	23B	S				
	4050 (1)		23B	S				

* The central profile is indicated with an asterisk; P, penicillin; E, erythromycin; CC, clindamycin; Te, tetracycline; SXT, cotrimoxazole; S, sensitive to all tested antibiotics; n, number of isolates; SLV, single-locus variant; DLV, double-locus variant.

## Data Availability

The data that support the findings of this study are available on request from the last author (I.K.-G.).

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
