# Peer review of "The Molecular Epidemiology of Pneumococcal Strains Isolated from the Nasopharynx of Preschool Children 3 Years after the Introduction of the PCV Vaccination Program in Poland"

_ijms, 2023, doi:10.3390/ijms24097883_

Round 1

Reviewer 1 Report

In the manuscript submitted by Kielbik and her colegues authors report on an outcome of analysis of 39 S. pneumoniae strains cultured from carriage in pre-school children in Poland three years after ten-valent vaccine implementation into the National Immunization Program.  These 39 isolates represent a subset of 41 S. pneumoniae  described by the same group in their paper cited as refrence number 20. Authors aimed to characterize the strains based on analysis of serotypes, antimicrobial susceptibility determined with phenotypic method(s) and molecular-method-based analysis of (selected) genetic determinants of resistance, genes coding for pneumococci pili and clonality based on allelic variation within housekeeping genes targeted in MLST.

Unfortunately, the reviewer finds the manuscript poorly written and cannot recommend it for the publication in the journal.

Examples of poor of English:

line 66: “Considering high recombination rates in pneumococci [14], determination of the genetic diversity and serotype relevance of pneumococcal strains, including antibiotic resistant strains, can be useful to control the prevalence of this bacterium in population.

Line 167: “Moreover, the diversity of circulating genotypes often masked by multiple capsular types result in complication of S. pneumoniae epidemiology. This indicates that the competition between different pneumococcal serotypes within the nasopharyngeal niche is a prerequisite for bacterial colonization.”

Line 175: The statement that “… 58,4% of isolate were belonging to one of 43 clones…” indicates that each of 43 clones was represented by at least a single pneumococcal isolate there.

Of note, the diversity can be measured. Authors claim (line 171) that “…MLST showed high diversity among isolates tested“.  What was the reference (presumably lower diversity) authors comparing  the results of their study to?

In absence of any statistical analysis, it is unknown if  differences in resistance patterns between this and previous study, and between vaccinated and un-vaccinated children are actually of any significance. Consequently, not only percentages but also absolute numbers need to be listed whenever there a difference (or lack of such) is reported.

According to line 48, presence of pili “favor (s) nasopharyngeal colonization and invasion.” Can authors cite any original paper(s), and not a review, reporting on piliated stains being over-represented among strains isolated from invasive pneumococcal disease? What if pilus would be over-represented among carried pneumococci? Also, in absence of any data on pili in pneumococci circulating in carriage in the same demographic group prior to vaccine introduction, a relevance of results reported in line 97-99 is obscure.   

Figure 1 seems redundant as it recycles the results already reported in Table 1.

Table 1: It is not clear what SLVs and DLVs reported in the third column mean? What “S”i n “Resistance Pattern” column stands for?

Figure 2 is very confusing. For one, all circles are the same size. What “182203” stands for? In number of instances it masks the number of loci between MLST profiles.

Line 43; all pneumococcal major autolysins are actually of choline-binding proteins.

Author Response

We would like to thank the Reviewer for the suggestions of revision, which undoubtedly improved our paper.

We have revised the manuscript in accordance with the comments and we enclose a revised manuscript with all changes highlighted in yellow. We introduced minor modifications in the Introduction, Results, Methods and the Discussion sections, based on reviewers’ comments.

Please find below detailed responses to each of the comments indicating what has been revised.

In the manuscript submitted by Kielbik and her colegues authors report on an outcome of analysis of 39 S. pneumoniae strains cultured from carriage in pre-school children in Poland three years after ten-valent vaccine implementation into the National Immunization Program.  These 39 isolates represent a subset of 41 S. pneumoniae  described by the same group in their paper cited as refrence number 20. Authors aimed to characterize the strains based on analysis of serotypes, antimicrobial susceptibility determined with phenotypic method(s) and molecular-method-based analysis of (selected) genetic determinants of resistance, genes coding for pneumococci pili and clonality based on allelic variation within housekeeping genes targeted in MLST.

Unfortunately, the reviewer finds the manuscript poorly written and cannot recommend it for the publication in the journal.

Re: Thank you for the critical opinion which persuade us to improve the manuscript. The manuscript has been corrected by English native speaker. We hope that in this form it is appropriate for publication.

Examples of poor of English:

line 66: “Considering high recombination rates in pneumococci [14], determination of the genetic diversity and serotype relevance of pneumococcal strains, including antibiotic resistant strains, can be useful to control the prevalence of this bacterium in population.”

Re: The sentence was rewritten (lines 70-73).

Line 167: “Moreover, the diversity of circulating genotypes often masked by multiple capsular types result in complication of S. pneumoniae epidemiology. This indicates that the competition between different pneumococcal serotypes within the nasopharyngeal niche is a prerequisite for bacterial colonization.”

Re: The sentences were corrected (lines 183-186).

Line 175: The statement that “… 58,4% of isolate were belonging to one of 43 clones…” indicates that each of 43 clones was represented by at least a single pneumococcal isolate there.

Re: The sentence was clarified (lines 194-196).

Of note, the diversity can be measured. Authors claim (line 171) that “…MLST showed high diversity among isolates tested“.  What was the reference (presumably lower diversity) authors comparing  the results of their study to?

Re: We have got your point. The discriminatory power of the MLST method was calculated using Simpson’s diversity index (SID). The manuscript has been modified accordingly (lines 188-189).

In absence of any statistical analysis, it is unknown if  differences in resistance patterns between this and previous study, and between vaccinated and un-vaccinated children are actually of any significance. Consequently, not only percentages but also absolute numbers need to be listed whenever there a difference (or lack of such) is reported.

Re: Thank you for the suggestion. We have added the missing numbers. Statistic analysis of difference between strains isolated from vaccinated and unvaccinated children was presented in our previous study. We added statistic analysis of differences in antibiotic resistance between isolates collected from vaccinated and unvaccinated children (lines 111-118).

According to line 48, presence of pili “favor (s) nasopharyngeal colonization and invasion.” Can authors cite any original paper(s), and not a review, reporting on piliated stains being over-represented among strains isolated from invasive pneumococcal disease? What if pilus would be over-represented among carried pneumococci? Also, in absence of any data on pili in pneumococci circulating in carriage in the same demographic group prior to vaccine introduction, a relevance of results reported in line 97-99 is obscure.  

Re: Indeed, paper by Fu et al. reported that the rlrA gene was identified in 22 of 37 (59.5%) isolates, and sipA was detected in 67.6% (25 isolates) of the pneumococcal isolates. (J. Fu, L. Li, Z. Liang, S. Xu, N. Lin, P. Qin, X. Ye, E. Mcgrath. Etiology of acute otitis media and phenotypic-molecular characterization of Streptococcus pneumoniae isolated from children in Liuzhou, China. BMC Infect. Dis., 19 (2019), p. 168). The explanation may be that 75.7% of the pneumococcal isolates in this study belonged to CC271 which is a multidrug resistant pneumococcal clone (Taiwan19F-14) described internationally. PI-1 and PI-2 were detected in 87% and 96% of these isolates. In our study, the high diversity of the STs was observed and I believe we can treat the data as genuine. However, taking into account the small number of isolates tested, a representativity is weak.

Figure 1 seems redundant as it recycles the results already reported in Table 1.

Re: We agree that the results are presented in Table 1, however, this chart is the aggregate picture of the antibiotic resistance of tested strains shown from different perspectives (the association of drug resistance with serotype).

Table 1: It is not clear what SLVs and DLVs reported in the third column mean? What “S”i n “Resistance Pattern” column stands for?

Re: We agree that third column is unnecessary. We have deleted it and explained “S” abbrev.

Figure 2 is very confusing. For one, all circles are the same size. What “182203” stands for? In number of instances it masks the number of loci between MLST profiles.

Re: Phylogenic tree visualization was performed with use of the PHYLOViZ on-line program to generate minimum spanning and neighbour joining trees. As was presented in Table 1, only ST  1349 was represented by 6 isolates with serotype 23B (it is visible on the tree). Rest of STs were represented mainly by 1 isolate and this is shown on the phylogenic tree. We have changed the nodes labels from ID numbers to ST numbers for clarification.

Line 43; all pneumococcal major autolysins are actually of choline-binding proteins.

Re: Thank you for the comment. We clarified the information (lines 46-48).

Reviewer 2 Report

I enjoyed reading this work, tthe authors presented data of molecular epidemiology of pneumococcal strains in Poland isolated from nasopharynx after introduction of vaccine program

Minor concerns: 

Line 20: “DLVs” can the authors change this to the full name since it’s the first time that is introduced.

Line 95: can the authors elaborate what this mean “tested antimicrobial agents in 43.6%.” seems like an unfinished sentence.

Line 96: can the authors change “1”… to “one”

Line 98: how did the authors determine expression of both pilus?

Line 140: how was capsular switching determined in Spain 9V-3 clone and variants to serotype 35B 140 and 6A, and in DLVs of Tennessee 23F-4 to serotype 23A? can the authors please make this clear.

Author Response

We would like to thank the Reviewer for the suggestions of revision, which undoubtedly improved our paper.

We have revised the manuscript in accordance with the comments and we enclose a revised manuscript with all changes highlighted in yellow. We introduced minor modifications in the Introduction, Methods Results and the Discussion sections, based on reviewers’ comments.

Please find below detailed responses to each of the comments indicating what has been revised.

Line 20: “DLVs” can the authors change this to the full name since it’s the first time that is introduced.

Re: It was corrected.

Line 95: can the authors elaborate what this mean “tested antimicrobial agents in 43.6%.” seems like an unfinished sentence.

Re: The sentence was clarified.

Line 96: can the authors change “1”… to “one”

Re: It was corrected.

Line 98: how did the authors determine expression of both pilus?

Re: Isolate was described as piliated with both PI-1 and PI-2, if PCR products were detected both for primers targeting rrgA (3’-end) and rrgB (5’-end) and those targeting pitA (3’-end) and pitB (5’-end) amplified PI-2. Primers targeting rrgA (3’-end) and rrgB (5’-end) amplified PI-1, while those targeting pitA (3’-end) and pitB (5’-end) amplified PI-2. To confirm the absence of these genes, we used primers that targeted the flanking regions. It was described in the Methods.

Line 140: how was capsular switching determined in Spain 9V-3 clone and variants to serotype 35B 140 and 6A, and in DLVs of Tennessee 23F-4 to serotype 23A? can the authors please make this clear.

Re: Thank you for pointing this out. We know that we should perform deeper genetic analysis (i.e.  WGS) for such a statement that these cases were capsular switching. We modified the text and clarified that capsular switching was quite likely.

Reviewer 3 Report

Title of article reviewed:  Molecular epidemiology of pneumococcal strains isolated from nasopharynx of preschool children 3 years after introduction of vaccination program in Poland

GENERAL COMMENTS: The whole work is well presented. There are only slight details to be corrected throughout the whole manuscript.

I am writing the following specific comments, of how I think the text should have been written correctly.

ABSTRACT

Line 21: DLV-Please change spelling.

Line 21: The main transposon ≠ the major transposon

Line 25: ..in CG4, consisting….

Line 26:…clone and expressing

INTRODUCTION

 Line 37: etiological ≠ etiologic

Line 38: bacteremia

Line 42: ..is armed with several virulence factors….

Line 50-51: …in Europe which introduced……mandatory vaccination…..

Line 60: In recent study, completed in 2020….

Line 64: …of PCVs is expected to cause serotype replacement and alter…

Line 69: Multilocus… Start the next paragraph with this sentence

RESULTS

Line 91: the 39 pneumococcal ….were identified by PCR detection of lytA and ply genes, whereas all of them were identified as capsular by detection of cpsA gene.

Line 93: They belonged to 10 different serotypes.

Line 94: 17 out of the 39 pneumococcal isolates (43.6%) were susceptible to all tested antimicrobial agents.

Line 95: serotype 23B

Line 103: Check clindamycin resistance frequency according to the figure

Line 103: All tetracycline resistant…start a new paragraph here, up to Table 1 (line108).

The main transposon…start a new paragraph.

Figure 1. There is no need for the numbers in the columns.

Please put them in a supplementary table.

Or, maybe better to change the graphic and show the resistance per serotype.

Line 128: (MLST) and PMEN clones

Line 129: Which STs were the new ones?

Line 130: genetic groups…Are they Clonal Groups (CGs)

MLST revealed 6 Clonal Groups-6 STs were associated….

Line 133: Eighteen…start a new paragraph

Line 144: Table 1

Lines 147-149 must be transferred in the beginning- (line 133) last sentences of the first paragraph

DISCUSSION

Line 165:….after introduction of vaccination….

Line 176: RTI???-What stands for?

Line 215: However…start a new paragraph

Line 223: Tn916 named…

Line 225: strain ≠ strains

MATERIALS AND METHODS

Add paragraph 4.4-line 284: Cluster analysis….

Title: Cluster analysis an PMEN database

COCNCLUSIONS

Lines 293-297: Transfer to line 290

Author Response

We would like to thank the Reviewer for the suggestions of revision, which undoubtedly improved our paper.

We have revised the manuscript in accordance with the comments and we enclose a revised manuscript with all changes highlighted in yellow. We introduced minor modifications in the Introduction, Methods, Results and the Discussion sections, based on reviewers’ comments.

Please find below detailed responses to each of the comments indicating what has been revised.

GENERAL COMMENTS: The whole work is well presented. There are only slight details to be corrected throughout the whole manuscript.

I am writing the following specific comments, of how I think the text should have been written correctly.

Re: Thank you for your help.

 ABSTRACT

Line 21: DLV-Please change spelling.

Line 21: The main transposon ≠ the major transposon

Line 25: ..in CG4, consisting….

Line 26:…clone and expressing

Re: They were corrected.

INTRODUCTION

 Line 37: etiological ≠ etiologic

Line 38: bacteremia

Line 42: ..is armed with several virulence factors….

Line 50-51: …in Europe which introduced……mandatory vaccination…..

Line 60: In recent study, completed in 2020….

Line 64: …of PCVs is expected to cause serotype replacement and alter…

Line 69: Multilocus… Start the next paragraph with this sentence

Re: Corrections were made.

RESULTS

Line 91: the 39 pneumococcal ….were identified by PCR detection of lytA and ply genes, whereas all of them were identified as capsular by detection of cpsA gene.

Line 93: They belonged to 10 different serotypes.

Line 94: 17 out of the 39 pneumococcal isolates (43.6%) were susceptible to all tested antimicrobial agents.

Line 95: serotype 23B

Line 103: Check clindamycin resistance frequency according to the figure

Line 103: All tetracycline resistant…start a new paragraph here, up to Table 1 (line108).

The main transposon…start a new paragraph.

Re: The corrections were made.

Figure 1. There is no need for the numbers in the columns. Please put them in a supplementary table. Or, maybe better to change the graphic and show the resistance per serotype.

Re: The Figure was modified.

Line 128: (MLST) and PMEN clones

Line 129: Which STs were the new ones?

Line 130: genetic groups…Are they Clonal Groups (CGs)

MLST revealed 6 Clonal Groups-6 STs were associated….

Line 133: Eighteen…start a new paragraph

Line 144: Table 1

Lines 147-149 must be transferred in the beginning- (line 133) last sentences of the first paragraph

Re: All corrections were made.

DISCUSSION

Line 165:….after introduction of vaccination….

Line 176: RTI???-What stands for?

Line 215: However…start a new paragraph

Line 223: Tn916 named…

Line 225: strain ≠ strains

Re: The corrections were made.

 MATERIALS AND METHODS

Add paragraph 4.4-line 284: Cluster analysis….

Title: Cluster analysis an PMEN database

Re: It was provided.

 COCNCLUSIONS

Lines 293-297: Transfer to line 290

Re: We believe that such sequence of information is more clear for the reader.